# Phase Equilibria in Ternary System CsBr-AgBr-InBr_3_

**DOI:** 10.3390/ma16020559

**Published:** 2023-01-06

**Authors:** Rustam K. Kamilov, Jahongir Z. Yuldoshev, Alexander V. Knotko, Anastasia V. Grigorieva

**Affiliations:** 1Department of Material Science, Lomonosov Moscow State University, 119991 Moscow, Russia; 2Department of Chemistry, Lomonosov Moscow State University, 119991 Moscow, Russia

**Keywords:** double perovskite, complex halides, ternary system, phase equilibria

## Abstract

The double perovskite halides A_2_B^I^B^III^X_6_ provide flexibility for various formulation adjustments and are of less toxicity in comparison with well-discussed complex lead halide derivatives. Such type of structure can be formed by replacing two Pb^2+^ ions in the cubic lattice with a pair of non-toxic heterovalent (monovalent and trivalent) metal cations, such as silver and indium. The aim of this work is to briefly characterize the phase equilibria in the ternary system CsBr-AgBr-InBr_3_ and investigate the thermodynamic availability of synthesis of Cs_2_AgInBr_6_ double perovskite phase by solid-state sintering or melt crystallization. The results demonstrate the unfeasibility of the Cs_2_AgInBr_6_ phase but high stability of the corresponding binary bromides perspective for optoelectronics.

## 1. Introduction

New materials for “green” energy technologies, including photovoltaics, have been of great importance over the past decades [1,2]. Metal halide perovskite photovoltaic systems have attracted interest from the scientific community and industry representatives. Interest was initiated by the work of Kojima et al. in 2006 using lead halide perovskite in a photoelectrochemical cell with a structure similar to colored solar cells, and later in 2009, it was published with an efficiency of 3.8% [3,4]. Solid-state perovskite solar cells developed by Kim et al. and Lee et al. in 2012 with efficiencies of 9.7% and 10.9%, respectively, became a breakthrough in the field of perovskite solar cells [5,6]. With each passing year, the efficiency of lead-based perovskite solar cells improved, which made them attractive materials all around the world. Nowadays, the efficiency of perovskite solar cells has exceeded 25% [7]. The rapid increase in efficiency is attributed to the excellent optoelectronic properties of lead halide-based perovskites, originated from a suitable forward bandgap, high absorption coefficient, long-range charge diffusion length, balanced electron and hole mobility, high dielectric constant, excellent carrier mobility and low binding energy of halide excitons. Complex halides with perovskite or perovskite-like structure have found application in optoelectronics, such as emitters in light-emitting diodes (LED) [8,9], lasers [10], materials for photodetectors [11,12].

Despite their impressive characteristics, serious disadvantages of the most efficient light harvesters with perovskite structure are their toxicity [13,14], poor resistance to heat [15,16], light [16,17], oxygen [17], and self-decomposition [18]. The presence of lead in the chemical composition of hybrid organo-inorganic perovskites of lead halides makes them toxic due to the easy dissolution of Pb^2+^ ions in water, which can cause environmental pollution harmful to humans and the ecosystem [15,17]. Researchers have made significant efforts to find less toxic and more stable alternative perovskites with good photophysical properties [19,20,21]. To reduce toxicity, efforts have been made in the past few years to replace lead with a divalent non-toxic metal cation while maintaining the perovskite structure. Sn and Ge appear to be the most suitable alternatives to Pb in the perovskite structure. It has been observed that non-toxic Sn-based halide perovskites decompose very quickly upon contact with air, mainly due to the Sn^2+^ oxidation state, which makes them less stable than Pb-based perovskites [22]. The low light absorption, low dielectric constant, and low optical conductivity of Ge-based lead-free perovskites are responsible for their low photovoltaic characteristics [23]. Materials in which other bivalent elements of the periodic table were used have also been tested as a substitute for lead in the perovskite structure. They also exhibit reduced optoelectronic properties due to the large band gap, high effective carrier masses, and low absorption [24,25].

Recent theoretical calculations show that the halide structure of double perovskite, A_2_B^I^B^III^X_6_, which can be formed by replacing two toxic Pb^2+^ ions in the crystal lattice with a pair of non-toxic heterovalent (monovalent and trivalent) metal cations, is a promising alternative for the implementation of high-performance, lead-free, and stable perovskite solar cells. Consequently, the crystal structure of double perovskite A_2_B^I^B^III^X_6_ deserves in-depth study since it provides the possibility of easier substitution and inclusion of various metal cations with different oxidation states in the B-position, various organic and inorganic ions in the A-position and variations in the halide composition in X-positions [26,27,28]. Halide double perovskites with Ag^+^ in the B^I^-position and a heavier halogen in the X-position can give a band gap suitable for photovoltaics [20,29].

Recently, a number of double perovskite compounds were synthesized [20,21,26]. At the same time, much more compounds with a double perovskite structure are predicted experimentally. Some of them would demonstrate a direct bandgap and high charge carrier mobility that is optimal for application in light-emitting diodes or photovoltaics. According to Wang et al. [30], the Cs_2_AgInBr_6_ phase has a direct bandgap of ~1.4 eV and excellent mobility of charge carriers that is promising for obtaining high-efficiency solar cells. Elsewhere, the authors found that Cs_2_AgInBr_6_ is stable mechanically [31]. According to Xiao et al. [29], in the ternary system CsBr-AgBr-InBr_3,_ the formation of binary bromides CsAgBr_2_ and Cs_3_In_2_Br_9_ is thermodynamically more favorable. Consequently, here, we present the most recent experimental results on the synthesis of the double perovskite Cs_2_AgInBr_6_ and also investigations of phase equilibria for the ternary CsBr-AgBr-InBr_3_ system using single or binary bromides as precursors.

## 2. Experimental

### 2.1. Analysis Methods

X-ray analysis of the samples was carried out using a Rigaku 2500 D-max diffractometer (Rigaku, Tokyo, Japan) with a proportional point detector on CuKα radiation (λ = 1.5418 Å) rotating copper anode in the range 2θ of 10–80° with a step of 0.02°. The obtained data were processed using WinXPow (STOE & Cie, Version 1.07, Darmstadt, Germany) and Jana2006 (ECA-SIG#3/Institute of Physics, Prague, Czech Republic) applications.

To determine the elemental composition of the obtained samples, the materials were examined by X-ray emission microanalysis using a Leo Supra 50 VP instrument (LEO Carl Zeiss SMT Ltd., Oberkochen, Germany) with an X/MAX X-ray energy dispersive detector (EDS) (Oxford Instruments, High Wycombe, UK) at a electron accelerating voltage of 21 kV.

The Raman spectra were studied on an InVia Raman Microscope spectrometer (Renishaw, New Mills, UK) equipped with an argon laser with a wavelength of 532 nm and a helium-neon (He-Ne) laser with a power of 20 mW with a wavelength of 632.8 nm and a spectral filter (5% of the total intensity). All spectra were obtained using a 50× objective by 100-fold signal accumulation, excitation time 1 s. A plate of (100)-oriented single-crystal silicon with a characteristic Raman mode of 521.5 cm^−1^ served as a standard sample for the preliminary calibration of the device.

Differential scanning calorimetry was carried out on a high-temperature differential scanning calorimeter DSC 404 C Pegasus (NETZSCH, Selb, Germany), at a temperature range of 200–400 °C in an argon atmosphere.

### 2.2. Synthesis of Samples in Ternary System CsBr-AgBr-InBr_3_

Solid-phase and heterophase ampoule synthesis was used as the main approach in the synthesis of the compositions of double perovskites, binary bromides, as well as the study of various binary sections in the composition of these ternary systems [2]. Cesium bromide was taken after Sigma-Aldrich (99.999%). Silver bromide was synthesized via an ion-exchange process using silver nitrate (“Reahim”, pure) and potassium bromide (“Rushim”, pure) as precursors. A light-yellow precipitate was washed out with MilliQ distilled water and heated at 60 °C in nitrogen atmosphere. Indium tribromide was synthesized by the solution method using elementary indium (“RedkyMetal.RF”, 99.999%) and hydrobromic acid (“Reahim”, analytical grade) [32,33]. We dissolve metallic indium in a solution of hydrobromic acid. Further, this solution is evaporated at 60 °C.

This approach allows one to study phase equilibria in a closed vessel. Samples of simple bromides were placed in quartz ampoules with an inner diameter of 8 mm, the total weight of the samples was 2 g. The ampoules were sealed after their evacuation (0.072 bar). At the same time, when placing the components of the system in ampoules and choosing the conditions for heat treatment of the ampoules, the differences in the volatility of the components were taken into account.

Several points from the CsBr-AgBr-InBr_3_ system were selected for analysis of two “double perovskites” predicted theoretically (Figure 1a).

According to the literature [22,29], the above-mentioned ternary systems contain phases with a double perovskite structure. Double perovskites of the composition Cs_2_AgInBr_6_ have a face-centered cubic structure with the space group Fm3m and with a lattice constant of about 11 Å. Unlike ABX_3_ perovskite, Cs_2_AgInBr_6_ consists of AgBr_6_ and InBr_6_ octahedra alternating along the [100], [010], and [001] directions. The B-site cations Ag^+^ and In^3+^ are formed in the form of an ordered structure of rock salt, as shown in Figure 1b.

Sample weights per 1 g of the finished sample are shown in Table 1 for the CsBr-AgBr-InBr_3_ system.

Samples of theoretically predicted composition Cs_2_AgInBr_6_ were synthesized by the solid-phase ampoule method. The conditions for annealing were T = 300–650 °C and then annealed at this temperature for 96 h, according to the conditions of solid-phase synthesis of similar compositions [34].

## 3. Results and Discussion

Point ‘1’ in CsBr-AgBr-InBr_3_ Gibbs’s triangle corresponds to a “double perovskite” composition Cs_2_AgInBr_6_. The synthesis was performed at different temperatures (300 °C—solid-phase synthesis, 450 °C—partially melted synthesis, 650 °C—crystallization from the melt) to reach the most optimal thermodynamic parameters for the perovskite phase formation. The samples obtained at 300°C and assumed as products of solid-phase synthesis are light grey-colored powders, while the samples annealed at 650 °C correspond to melt crystallization and are “single-piece” and dark-grey colored.

The XRD pattern of the “double perovskite” Cs_2_AgInBr_6_ sample, annealed at 300 °C demonstrates some reflections at the same 2Θ value as for double halides, namely, Cs_2_AgBr_3_ (PDF#01-072-9840) and Cs_3_In_2_Br_9_ (Springer materials ID: sd_1712349). Similar results were reported recently by Fan et al. [35], who discussed the vapor-deposited Cs_2_AgBiBr_6_ thin films.

Three binary systems are parts of the CsBr-AgBr-InBr_3_ ternary system. In the CsBr–AgBr binary system, there are two phases of cesium bromoargenates(I) CsAgBr_2_ and Cs_2_AgBr_3_ [36], and in the CsBr–InBr_3_ system, three binary cesium bromoindates(III) bromides, namely, Cs_3_In_2_Br_9_ [37], Cs_3_InBr_6_, and Cs_2_InBr_5_ [38] are formed. In the ternary diagram, the ‘double perovskite’ phase belongs to two double sections, including AgBr—Cs_2_InBr_5_ and InBr_3_—Cs_2_AgBr_3_.

According to the phase XRD analysis performed (Figure 2), the Cs_2_AgInBr_6_ phase was not formed, but the sample ‘1’, annealed at 300 °C included two phases of binary bromides Cs_3_In_2_Br_9,_ Cs_2_AgBr_3_. Samples ‘1’, annealed at 450 and 650 °C include the phases Cs_3_In_2_Br_9,_ Cs_2_AgBr_3_, and Cs_2_InBr_5_. Additionally, some reflections of the AgBr phase would be found, which are not unique only for this phase but overlap with the reflections of the phases of binary bromides, namely, Cs_3_In_2_Br_9_ and Cs_2_AgBr_3_.

The reflections of the Cs_2_InBr_5_ phase corresponded to the hydrate form Cs_2_InBr_5_·H_2_O [39] which is not shown in the phase diagram reported earlier by Dudareva et al. [38]. This may be due to the fact that X-ray phase analysis occurred in open air and there were water vapors in air. No reflections of the unhydrous Cs_2_InBr_5_ phase were found in XRD data for other samples in the ternary system CsBr-AgBr-InBr_3_. The hydration reaction proceeds at room temperature [39] according to the Equation (1):Cs_3_In_2_Br_9_ + CsBr + 2H_2_O = 2Cs_2_InBr_5_∙H_2_O(1)

In order to refine the stoichiometry in the course of ampoule synthesis, some of the samples were additionally studied by the EDX method. Figure 3 and Table 2 show the results of EDX spectroscopy of the samples synthesized by the ampoule method at 300 °C and 650 °C, respectively. It can be assumed that for the samples obtained by the ampoule synthesis, deviations from theoretical values can be associated both with the uneven distribution of indium and silver in an ampoule during the heat treatment and also with various morphology of the crystallites of the resulting phases, i.e. limitations of the EDS method (Figure 3).

The Raman spectra of the samples synthesized at 300 °C and 650 °C (Figure 4) contain intense vibrational modes corresponding to the Cs_3_In_2_Br_9_ phase: ν (A_1g_)—165 cm^−1^, ν (E_g_)—216 cm^−1^ и 111 cm^−1^ [39]. There are also two “shoulders” at 143 and 210 cm^−1^, presumably, corresponding to the Raman modes of the Cs_2_InBr_5_ phase [40]_._ It is remarkable that the crystalline phase Cs_3_In_2_Br_9_ is not shown in the binary phase diagram CsBr—reported previously by Dudareva et al. [38]. The Raman spectrum of the composition ‘1’ annealed at 450 °C corresponds well to a superposition of the two spectra discussed above (Figure 4).

To study the phase equilibria in the ternary system of bromides, a number of additional compositions were synthesized and studied for its phase analysis.

A Cs_3_In_2_Br_9_ sample ‘3’ obtained at a temperature of 300 °C and an annealing time of 96 h according to the XRD analysis (Figure 5) is a single-phase sample of the desired phase Cs_3_In_2_Br_9_ (Springer materials ID: sd_1712349). The Raman data also correspond well to the spectrum of Cs_3_In_2_Br_9_ phase described by Zhou et al. [39].

Samples of the CsAgBr_2_ binary bromide (point ‘2’) obtained at a temperature of 260 °C and an annealing time of 96 h were also single-phase, according to the corresponding XRD data (Figure 6a). The typical Raman spectrum of caesium dibromoargenate(I) CsAgBr_2_ is given in Figure 6b. The spectrum includes two maxima, namely, the strongest one at 155 cm^−1^ and less intensive at 124 cm^−1^ that, most likely, correspond to vibrational models ν_1_ and δ of [AgBr_4_]^3−^ vibrations [41].

In order to get more information about the named above binary phase equilibria and the CsBr-AgBr-InBr_3_ ternary system, three additional compositions were synthesized at three intersections, namely, the CsAgBr_2_-InBr_3_ and AgBr-Cs_3_InBr_6_ sections (point ‘4’), CsAgBr_2_-InBr_3_ and AgBr-Cs_3_In_2_Br_9_ sections (point ‘5’), as well as CsAgBr_2_-Cs_3_In_2_Br_9_ and AgBr-Cs_2_InBr_5_ (point ‘6’). The annealing temperature in all cases was 200 °C to avoid evaporation of volatile indium(III) bromide from the reaction zone in an ampoule.

The XRD patterns of samples ‘4’, ‘5’, and ‘6’ are given in Figure 7, Figure 8 and Figure 9. Identification of reflections in diffractograms was performed using the XRD data to the stable and metastable phased known for the binary systems (see Figure 1a). No new phases of ternary bromides were found. The diffraction patterns showed the presence of reflections of binary bromides as CsAgBr_2_ (PDF-2 (38–850)), Cs_3_In_2_Br_9_ (Springer materials ID: sd_1712349), and a Cs_3_InBr_6_ (Materials projects ID:mp-1112651), as well as a precursor AgBr (PDF-2 (79–149)). We found no reflections of the less volatile crystalline CsBr that makes an assumption of having a stable quasi-binary cross-section Cs_2_AgBr_3_—Cs_3_In_2_Br_9_.

Point ‘4’ is on the intersection of incisions of CsAgBr_2_-InBr_3_ and AgBr-Cs_3_In_2_Br_9_. The corresponding XRD data (Figure 7) showed the presence of strong reflections of the AgBr and also sharp and strong reflections of other two binary bromides of cesium bromoargenate(I) CsAgBr_2_ (PDF-2 (79–149)) and a single volatile bromide InBr_3_ (PDF-2 (79–281)). No reflections of cesium bromoindates(III) were found. Most likely, this means that the binary system AgBr-Cs_3_In_2_Br_9_ is not stable, while the CsAgBr_2_-InBr_3_ section is a thermodynamically stable cross-section.

Point ‘5’ is on the intersection of AgBr-Cs_2_InBr_5_ and CsAgBr_2_-InBr_3_. The corresponding XRD data (Figure 8) showed the presence of reflections of silver bromide AgBr, caesium dibromoargenate(I) CsAgBr_2_, and indium bromide InBr_3_ phases. This means that the binary system AgBr-Cs_2_InBr_5_ is metastable while CsAgBr_2_-InBr_3_ is a quasi-binary one.

Point ‘6’ is on the intersection of three incisions of CsAgBr_2_-Cs_3_In_2_Br_9_, AgBr-Cs_3_InBr_6_, and Cs_2_AgBr_3_-InBr_3_. The corresponding XRD data (Figure 10) showed the presence of reflections of CsAgBr_2_ (PDF-2 (79–149)), Cs_2_AgBr_3_ (PDF#01-072-9840), Cs_3_In_2_Br_9_ (Springer materials ID: sd_1712349), and InBr_3_ (PDF-2 (79–281)) phases. This means that the binary system AgBr-Cs_3_InBr_6_ is not stable, while CsAgBr_2_-Cs_3_In_2_Br_9_ and Cs_2_AgBr_3_-InBr_3_ would have thermodynamically stable cross-sections.

Figure 9 shows DSC data for a sample ‘6’ obtained by mixing of CsAgBr_2_ and Cs_3_In_2_Br_9_ binary bromides in stoichiometric amounts. It is shown that the heating curve has four endothermic minima. The last one at 372 °C (645 K) is the strongest and, most likely, corresponds to a liquidus. At lower temperatures, the minimum at 234 °C (507 K) is close to the eutectic temperature in the AgBr—InBr_3_ binary system. The nets minimum at 290 °C (563 K) is above the melting temperature of CsAgBr_2_ (270 °C) [36] and is close to the value of the eutectic temperature for the CsBr—InBr_3_ binary system. The third endothermic effect at 328 °C (601 K) could be associated with the melting of the Cs_3_In_2_Br_9_ phase, which is not present in the CsBr—InBr_3_ phase diagram reported elsewhere [42].

The cooling curve differs from the heating process. At higher temperatures, there is a smaller maximum at 380 °C (653 K) and the following narrower and sharper maximum at 372 °C (645 K). Most likely, such exothermic effects correspond to crystallization of Cs_3_In_2_Br_9_ phase and crystallization of indium(III) bromide (420 °C, 693 K). Then, there is a very weak maximum at 356 °C (629 K).

At lower temperatures, two crystallization maxima are observed at 293 °C (566 K) and 287 °C (551 K). Probably, the processes correspond to crystallization of Cs_2_AgBr_3_ and CsAgBr_2_ (270 °C). Moreover, below 250 °C, no crystallization of binary eutectics is found, as observed for the AgBr—InBr_3_ binary system at the heating process. Thus, it can be assumed that the phase composition of the sample after DSC measurement differs from the initial one, which may be a consequence of the reaction proceeding, for example, according to reaction 1.

Thus, the summary of XRD and DSC results for point ‘7’ indicates the possibility of solid-phase process (2) at 200 °C:2CsAgBr_2_ + 2Cs_3_In_2_Br_9_ = 4Cs_2_AgBr_3_ + 4InBr_3_(2)

The possibility of the process (2) contradicts the hypothesis about the quasi-binary characteristics of the section CsAgBr_2_—Cs_3_In_2_Br_9_ that originated from the instability of the double perovskite phase Cs_2_AgInBr_6_. At the same time, process (2) could take place in an open system and be negligible in closed systems, such as in ampoules.

Thus, a brief investigation of phase equilibria in the ternary system CsBr-AgBr-InBr_3_ gave some controversial results. Nevertheless, the phase analysis of the compositions demonstrated higher thermodynamic stability of three quasi-binary intersections, namely, Cs_3_In_2_Br_9_—Cs_2_AgBr_3_, Cs_2_AgBr_3_—InBr_3_, CsAgBr_2_—InBr_3_. The stability of the first one led to the full chemical transformation of cesium bromide as a precursor for the “double perovskite” composition in point ‘1’. The last two quasi-binary sections resulted from the high volatility of indium(III) bromide.

## 4. Conclusions

Investigation of the phase equilibria in the ternary system CsBr-AgBr-InBr_3_ is a brief one but gives important information on the processes that take part in the system at sub-solidus temperature region and above the melting temperatures. The proposed synthesis approaches for the estimated double perovskite phase by solid-phase synthesis and crystallization from the melt were not enough, and the Cs_2_AgInBr_6_ phase is unfeasible. The high stability of the phases of the binary bromides Cs_3_In_2_Br_9_ and Cs_2_AgBr_3_ prevents the formation of the double perovskite phase. Analysis of the ternary system CsBr-AgBr-InBr_3_ demonstrates three quasi-binary sections, namely, Cs_3_In_2_Br_9_—Cs_2_AgBr_3_, Cs_2_AgBr_3_—InBr_3_ and CsAgBr_2_—InBr_3_, which would be important for the synthesis of binary bromides and investigation of solid solutions in the ternary system or its sections.

## Figures and Tables

**Figure 1 materials-16-00559-f001:**
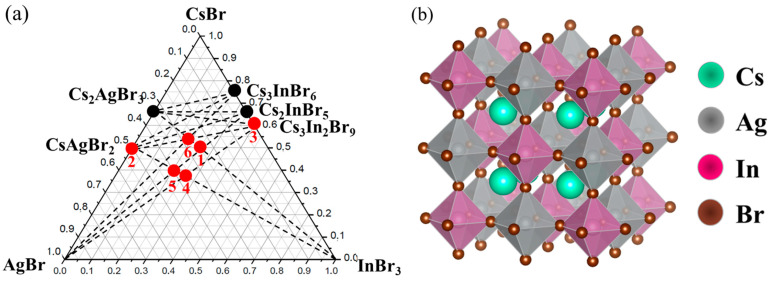
Phase triangle for the (**a**) CsBr-AgBr-InBr_3_ and (**b**) CsBr-AgBr-InBr_3_ ternary systems with selected compositions marked as points.

**Figure 2 materials-16-00559-f002:**
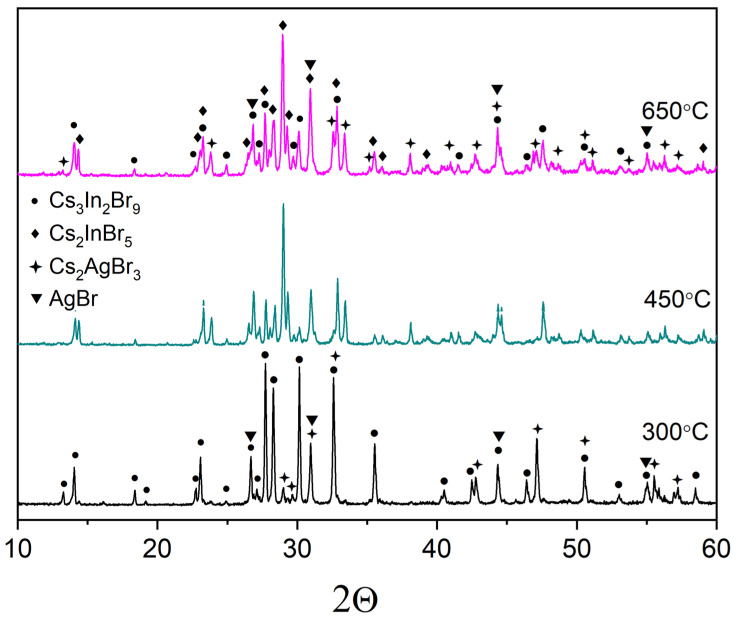
The diffraction pattern for the theoretical composition Cs_2_AgInBr_6_ (•—Cs_3_In_2_Br_9_ [Springer materials ID: sd_1712349]; ♦—Cs_2_InBr_5_ [37]; +—Cs_2_AgBr_3_ [PDF #01-072-9840]; ▼—AgBr (PDF-2 #79-149).

**Figure 3 materials-16-00559-f003:**
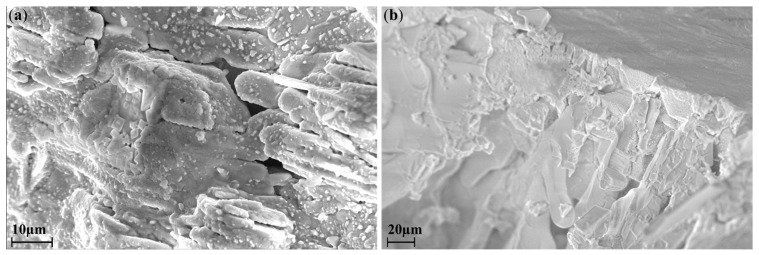
SEM images of samples obtained by the ampoule method at (**a**) 300 °C (**b**) 650 °C.

**Figure 4 materials-16-00559-f004:**
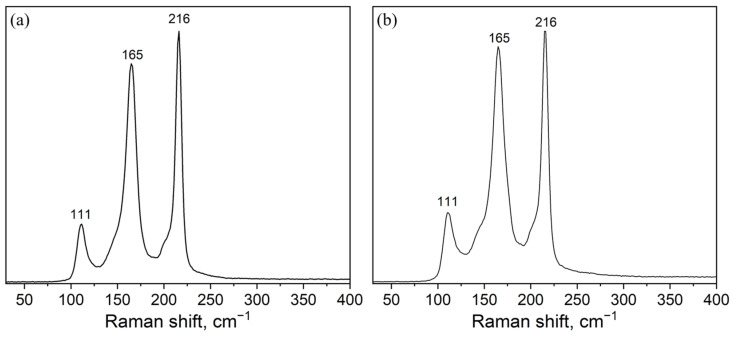
Raman spectra of samples of the theoretical composition Cs_2_AgInBr_6_ obtained by the ampoule method at (**a**) 300 °C (**b**) 650 °C. Laser excitation 532 nm.

**Figure 5 materials-16-00559-f005:**
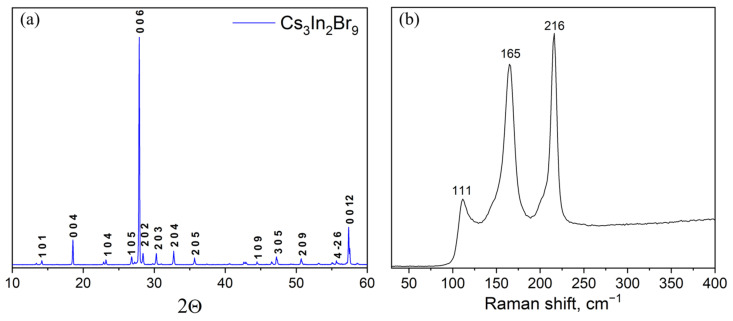
(**a**) XRD data and (**b**) characteristic Raman spectrum of a sample of general composition ‘3’ (Cs_3_In_2_Br_9_ phase).

**Figure 6 materials-16-00559-f006:**
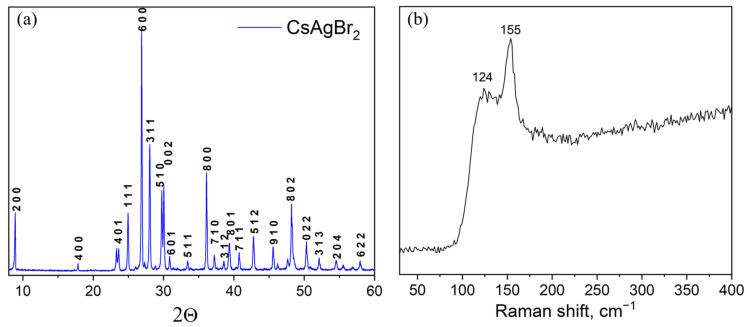
(**a**) XRD data and (**b**) Raman spectrum of a sample ‘2’ (CsAgBr_2_ phase).

**Figure 7 materials-16-00559-f007:**
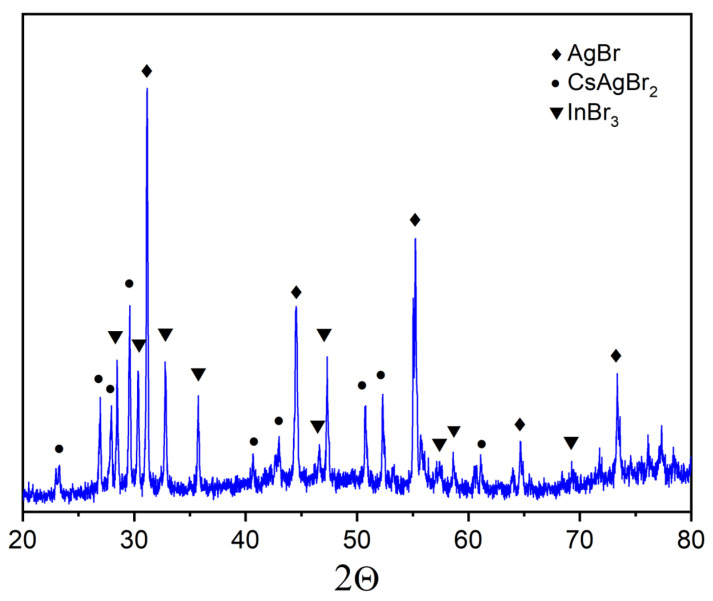
XRD data for point ‘4’ (intersection of incisions CsAgBr_2_-InBr_3_ and AgBr-Cs_3_In_2_Br_9_) (♦—AgBr PDF-2 (79–149); •—CsAgBr_2_ PDF-2 (38–850); ▼—InBr_3_ PDF-2 (79–281)).

**Figure 8 materials-16-00559-f008:**
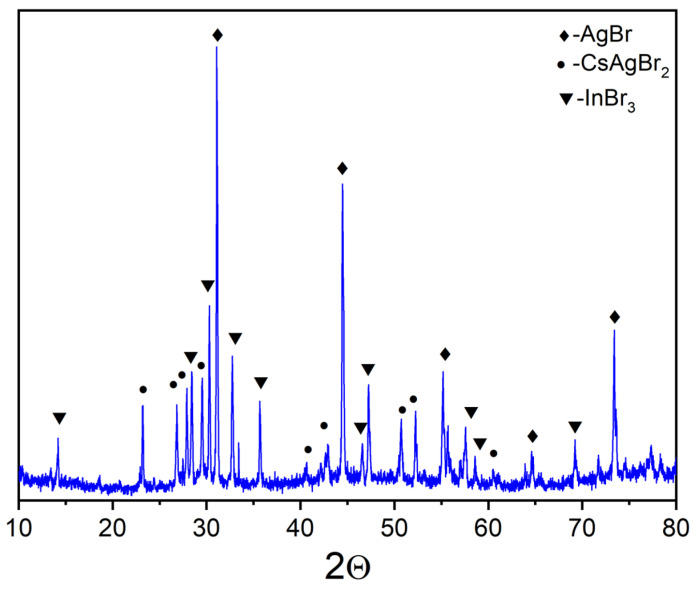
XRD data for point ‘5’ (intersection of incisions AgBr-Cs_2_InBr_5_ and CsAgBr_2_-InBr_3_) (♦—AgBr PDF-2 (79–149); •—CsAgBr_2_ PDF-2 (38–850); ▼—InBr_3_ PDF-2 (79–281)).

**Figure 9 materials-16-00559-f009:**
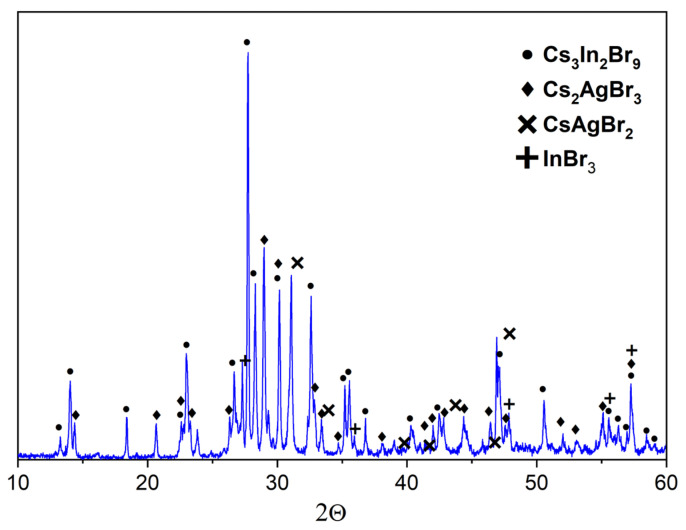
XRD data for point ‘6’ (intersection of incisions CsAgBr_2_-Cs_3_In_2_Br_9_ and AgBr-Cs_3_InBr_6_) (•—Cs_3_In_2_Br_9_ [Springer materials ID: sd_1712349]; ♦—Cs_2_AgBr_3_ (PDF#01-072-9840); ✕—CsAgBr_2_ PDF-2 (38–850); +—InBr_3_ PDF-2 (79–281)).

**Figure 10 materials-16-00559-f010:**
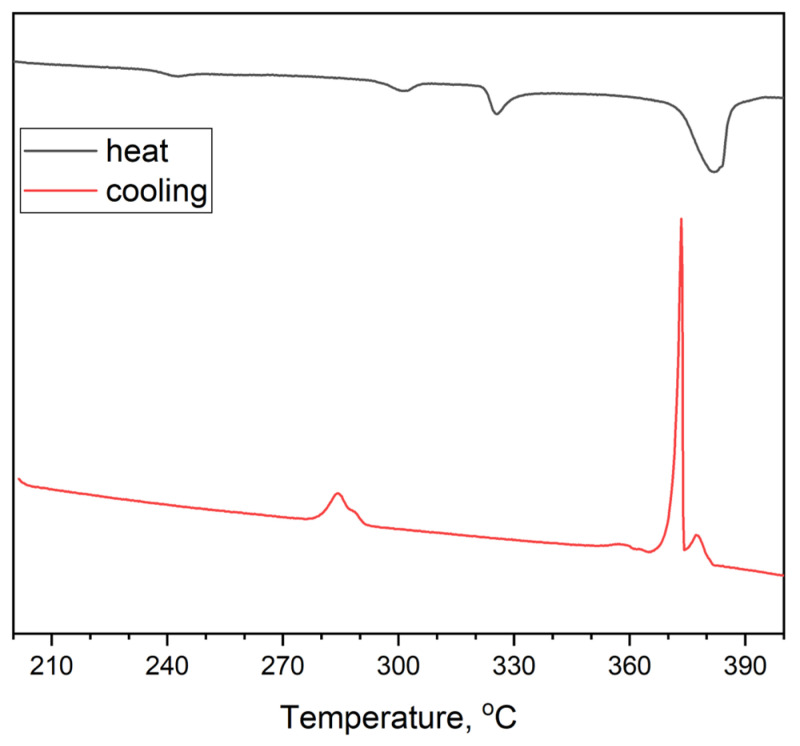
DSC data for the composition of point 6 at the intersection of the sections Cs_3_InBr_6_-AgBr, CsAgBr_2_-Cs_3_In_2_Br_9_, and Cs_2_AgBr_3_-InBr_3_, obtained by displacement of binary bromides CsAgBr_2_ and Cs_3_In_2_Br_9_.

**Table 1 materials-16-00559-t001:** Weighed weights for samples of the CsBr-AgBr-InBr_3_ system, calculated per 1 g of the final product.

Sample	Mole Fraction of Precursors			Weight of Precursors per 1 g. of Product		
	n (CsBr)	n (AgBr)	n (InBr_3_)	m (CsBr)	m (AgBr)	m (InBr_3_)
Point 1	0.5	0.25	0.25	0.44	0.19	0.37
Point 2	0.5	0.5	-	0.531	0.469	-
Point 3	0.6	-	0.4	0.474	-	0.526
Point 4	0.375	0.375	0.251	0.334	0.295	0.372
Point 5	0.546	0.273	0.181	0.501	0.221	0.278
Point 6	0.4	0.4	0.2	0.368	0.325	0.307

**Table 2 materials-16-00559-t002:** The atomic percentages of the elemental composition of the samples obtained by the ampoule method at T = 300 °C и 650 °C.

Synthesis Temperature, °C	Cs	Ag	In	Br	Cation Ratio Ag/In
T = 300 °C (solid-phase synthesis)	25	13	11	51	1.17
T = 650 °C (crystallization from the melt)	25	12	11	52	1.08

## Data Availability

Not applicable.

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
