# Peer review of "Phase Equilibria in Ternary System CsBr-AgBr-InBr3"

_materials, 2023, doi:10.3390/ma16020559_

Round 1

Reviewer 1 Report

Metal halide perovskite photovoltaic systems are of great interest as the world moves toward a green future. However, a current shortcoming is concerns over toxicity. A strategy to decrease toxicity is to replace Pb. Here the authors investigate the phase behavior of a potential double perovskite system. Namely, a system that had been predicted previously from theoretical calculations.

I do not have any criticisms. The work is straightforward. The experiments are well described, and the results are clearly presented. The work is not novel from a theoretical perspective, but is important. Experiments were performed, and observations were made. The main result is that the theoretically proposed double pervoskite is note feasible.

If I could offer any suggests, it would be to carefully consider revising the introduction and conclusion. With regards to the introduction, at the end I would include the major finding that the theoretically proposed double pervoskite is not feasible. Moreover, please explain the theoretical predictions in greater detail. I would also note that theory is not enough for the design of novel materials.... to solve our problems. Experiments play an essential role.

With regards to the conclusion, in its current form it is brief. I would consider expanding. You might consider emphasizing the significant of this work, and how it might impact future studies. I would again encourage you to emphasize the importance of experiments in the design process.

Author Response

This work concerns the experimental synthesis of the predicted double perovskite phase Cs2AgInBr6. The theoretical bandgap for this phase is 1.4 eV that makes the phase attractive as a light harvester for solid-state perovskite solar cells. We showed that the double perovskite phase is not feasible.

On the other hand, the experiments revealed crystallization processes occured in the ternary system in a temperature range of 300 - 650°C, including the formation of stable double bromides.

Both Introduction and Conclusion sections of the manuscript have been improved according to the comments.

Reviewer 2 Report

The authors explored the phase equilibria in ternary system CsBr-AgBr-InBr3 and thermodynamic availability of synthesis of Cs2AgInBr6 double perovskite phase by solid-state sintering or melt crystallization. The problem is very important in the new materials research and it is upto date. The authors clearly explained the procedures.  The results are useful and they provide some new insight in this research area. The paper is suitable for publication after incorporating the following comments.

Comments:

1)          Give the clear motivation of the present study on synthesis of the double perovskite Cs2AgInBr6.

2)          What is the specific reason to take the three values of Temperatures 300°C, 450°C, and 650°C.

3)          Why do no single bromides found as a result of full transformation to binary bromides Cs3In2Br9, Cs2InBr5 and Cs2AgBr3?.

4)          The authors took the Raman spectra of samples at 300°C, and 650°C. There is no spectra at 450°C? explain.

5)          Conclusion should be concise and brief.

6)          Check the language and format of the paper.

Author Response

1)          Give the clear motivation of the present study on synthesis of the double perovskite Cs2AgInBr6.

The synthesis of the double perovskite Cs2AgInBr6 is interesting due to the predicted direct bandgap of ~1.4 eV and excellent transport properties of charge carriers. Such characteristics are promising for high-efficient perovskite solar cells. We studied the possibility of formation of a double perovskite phase by a solid-phase interaction of precursors at 300°C, by crystallization from a melt at 650°C, and also at a medium temperature of 450°C.  The last one is above most of melting temperatures of the single and double bromides in the ternary system.

2)          What is the specific reason to take the three values of Temperatures 300°C, 450°C, and 650°C.

The choice of annealing temperatures of 300°C, 450°C and 650°C originated from information about melting temperatures of the corresponding single and double bromides. These temperatures are for solid-phase interaction (solid-phase sintering of single bromides at 300°C), partially melt (sintering of solid phases in the melt) and completely melt at 650°C (synthesis from the melt for all phases).

3)          Why do no single bromides found as a result of full transformation to binary bromides Cs3In2Br9, Cs2InBr5 and Cs2AgBr3?

In a new version of the Figure 2 we indicated additionally reflections of silver bromide AgBr. All XRD reflections which belong to the AgBr phase overlap with reflections of other detected phases, namely, Cs3In2Br9 and Cs2AgBr3. The XRD refinement with silver bromide phase shows R-factor of 0.2189 while without AgBr R is 0.2197. The corresponding part of the text was rewritten.

4)          The authors took the Raman spectra of samples at 300°C, and 650°C. There is no spectra at 450°C? explain.

The Raman spectrum for the sample synthesized at 450°C (partial melt synthesis) was not presented in the manuscript because two Raman spectra taken for the samples annealed at 300°C and 650°C (fully solid phase and fully melt synthesis, respectively) are identical.

5)          Conclusion should be concise and brief.

The conclusions are improved according to the recommendations of reviewers.

6)          Check the language and format of the paper.

The text of the article was improved in accordance with the template.